# Positive Prediction Value of Retinal Artery Occlusion Diagnoses in the Danish National Patient Registry: A Validation Study

**DOI:** 10.3390/jpm13060970

**Published:** 2023-06-08

**Authors:** Marie Ørskov, Tobias Primdahl Holst Nissen, Henrik Vorum, Torben Bjerregaard Larsen, Flemming Skjøth

**Affiliations:** 1Department of Cardiology, Aalborg University Hospital, 9000 Aalborg, Denmark; 2Aalborg Thrombosis Research Unit, Department of Clinical Medicine, Faculty of Health, Aalborg University, 9000 Aalborg, Denmark; 3Department of Ophthalmology, Aalborg University Hospital, 9000 Aalborg, Denmark; 4Steno Diabetes Center North Jutland, 9000 Aalborg, Denmark; 5Unit for Clinical Biostatistics, Aalborg University Hospital, 9000 Aalborg, Denmark

**Keywords:** epidemiology, Danish registry, retina, retinal artery occlusion, validation

## Abstract

Purpose: The hospital registration of retinal artery occlusions in the Danish National Patient Registry has not previously been validated. In this study, the diagnosis codes were validated to ensure the diagnoses had an acceptable validity for research. The validation was performed both for the overall diagnosis population and at the subtype diagnosis level. Methods: The medical records for all patients with retinal artery occlusion with an incident hospital record in the years 2017–2019 in Northern Jutland (Denmark) were assessed in this population-based validation study. Furthermore, fundus images and two-person verification were assessed for the included patients when available. The positive prediction values for the overall diagnosis of retinal artery occlusion, as well as for the central or branch subtypes, were calculated. Results: A total of 102 medical records were available for review. The overall positive prediction value for a retinal artery occlusion diagnosis was 79.4% (95% CI: 70.6–86.1%), while the overall positive prediction value at the subtype diagnosis level was 69.6% (95% CI: 60.1–77.7%), with 73.3% (95% CI: 58.1–85.4%) for branch retinal artery occlusion and 71.2% (95% CI: 56.9–82.9%) for central retinal artery occlusion. For the stratified analyses at the subtype diagnosis, age, sex, diagnosis year, and primary or secondary diagnosis, the positive prediction values ranged from 73.5 to 91.7%. In the stratified analyses at the subtype level, the positive prediction values ranged from 63.3 to 83.3%. The differences among the positive prediction values of the individual strata of both analyses were not statistically significant. Conclusions: the validities of the retinal artery occlusion and subtype level diagnoses are comparable to other validated diagnoses and considered acceptable for use in research.

## 1. Introduction

Retinal artery occlusions cause monocular visual impairment and blindness [1]. Retinal artery occlusion is categorized according to their localization, including branch retinal artery occlusion, central retinal artery occlusion, and cilioretinal artery occlusion as the main subtypes [2]. Another subtype is transient retinal artery occlusion, where the occlusion dissolves and visual loss may be regained. The incidence of central retinal artery occlusion is 1.8–1.9 per 100,000 person years, increasing with age [3,4]. The low incidence rate makes it difficult to obtain large study populations when conducting research. In such situations, long-term longitudinal data from registries have proven to be an invaluable source, enabling the attainment of larger study populations.

The main cause of retinal artery occlusion is thromboembolism, in which the embolism may originate from the carotid artery, the heart, or the aortic arch [1]. Ophthalmic risk factors have been associated with the development of retinal artery occlusion as well, including glaucoma and cataract, which may suggest other pathways affecting the risk of developing retinal artery occlusion in addition to thromboembolism [5]. There may be differences in the pathophysiology of retinal artery occlusion for patients dependent on their age [1,6]. There are still questions that remain to be investigated concerning the more detailed pathophysiology for retinal artery occlusion. For instance, no available treatment has been confirmed in randomized clinical trials as an effective treatment for retinal artery occlusion, with better results than placebo [7]. Several treatment options have been investigated for retinal artery occlusion, including ocular massage, carbogen inhalation, intraocular pressure reducing drugs, carbonic anhydrase inhibitors, and hemodilution. However, these treatment options were identified as ineffective or, in some cases, even harmful [8,9]. Retinal artery occlusions are associated with several cardiovascular diseases and have previously been described as an equivalent to stroke [1,5,10,11]; however, retinal artery occlusion has not been investigated to the same extent as a stroke, although some differences in the risk factors for retinal artery occlusion and stroke have been identified indicating that the two diseases may not be equivalent [12]. Combined, all these uncertainties and differences need further investigation to understand retinal artery occlusion in more detail. Especially, further research is needed to investigate retinal artery occlusion and gain knowledge of the disease to help identify an effective treatment. Several of these questions can be addressed using observational studies and registry data, where large study populations can be ensured.

In Denmark, registries with long-term longitudinal health data constitute an important research facility. An advantage of registry-based studies is that the data already exist, which saves the researcher a substantial amount of time in terms of data collection. Furthermore, a study’s independent registration of data minimizes the risk of selection bias [13]. However, the validity of the data is a great concern when working with registry data. The registration needs to be accurate for the data to be useful. While validation studies have primarily focused on prevalent diseases, it is important to investigate the validity for all diagnosis codes employed in research. This comprehensive approach is crucial, as low positive predictive values associated with specific diagnoses have the potential to significantly influence the study outcomes. Therefore, thorough validation across a wide range of diseases is essential to ensure the accuracy and reliability of research findings. Therefore, the aim of this study was to validate the positive prediction value for retinal artery occlusion diagnoses assessed at a specialized ophthalmologic department at a public hospital in Denmark. Data were validated both according to the overall diagnosis of retinal artery occlusion and on a subtype diagnosis level (including branch retinal artery occlusion and central retinal artery occlusion).

## 2. Materials and Methods

### 2.1. Setting

The Danish healthcare system is mainly supported through a progressive national income tax that ensures free, accessible healthcare for all. Denmark is divided into five regions, each responsible for healthcare in their respective region but regulated by national authorities to ensure uniform healthcare service throughout the country. This study used data from the North Denmark Region. The number of citizens in the North Denmark Region in 2022 was approximately 592,000, which was about 10% of the Danish population [14]. Patients with acute eye disease or the need for elective treatment visit a single public ophthalmologic department in the region: Department of Ophthalmology, Aalborg University Hospital. Public health insurance patients are usually referred to the university department by ophthalmologists working in their own clinics located in primary care. Public health insurance in Denmark ensures coverage for all permanent residents of the country. It is worth noting that patients diagnosed in a private hospital typically transition to receiving treatment in a public hospital setting.

### 2.2. Registry

Data from the Danish healthcare system is collected for administrative purposes and for quality control in several national databases. These databases are also available for research, and all information obtained from the registries can be linked using the personal identification number provided to all permanent residents of Denmark at birth or migration [15]. The main registry with data on diagnoses is the Danish National Patient Registry. In this registry, all inpatient and outpatient diagnoses have been registered since 1977, with additional information added subsequently [16,17]. A patient is discharged with one primary discharge diagnosis (A diagnosis) and up to 20 secondary discharge diagnoses (B diagnoses). Since 1993, data have been registered using the 10th revision of WHO’s International Statistical Classification of Diseases and Related Health Problems (ICD-10); before 1993, the 8th revision was used [16,17].

### 2.3. Study Population

The included study population consisted of all patients with a primary or secondary first-time discharge diagnosis of retinal artery occlusion from the ophthalmologic department at Aalborg University Hospital in the North Denmark Region from the period 1 January 2017 to 31 December 2019. Both inpatient and outpatient diagnoses were included. The following ICD-10 codes for retinal artery occlusion were included: transient retinal artery occlusion—DH340 and DH340A; central retinal artery occlusion—DH341, DH341A, and DH341B; and branch retinal artery occlusion—DH342, DH342A, DH342B, and DH342C. The population was validated overall, where all ICD-10 codes for retinal artery occlusion were included. Furthermore, we validated the subtypes of retinal artery occlusion, including branch retinal artery occlusion and central retinal artery occlusion. Because of the low prevalence and inadequate coding of other subtypes, we excluded the transient retinal artery occlusion registrations. The specific ICD-10 codes and the description for each are defined in Table 1. These represent the data also available from the Danish National Patient Registry.

### 2.4. Electronic Medical Record Review with Imaging Verification

Using the personal identification number provided to all residents of Denmark, we linked the electronic medical record to the individual patients with retinal artery occlusion identified in the Danish National Patient Registry.

The electronic medical records were reviewed by two researchers, and in the case of disagreement or uncertainty, an independent third researcher was asked to review the record. In the case of disagreement, all researchers discussed to reach an agreement.

The electronic medical record was reviewed for all included patients to determine whether the information in the electronic medical record matched the diagnosis registered for the patient. We started by identifying the record for each patient using the personal identification number. Journal entries from the ophthalmologic department were considered sufficient to verify the diagnosis, since intraocular examinations are only performed at ophthalmologic departments. When reviewing the medical records, we noted descriptions, visus, and symptoms and information regarding ICD codes, retinal artery occlusion subtype, diagnosis year, and diagnosis type (primary or secondary), as well as age and sex.

In addition to the assessment of the journal entry concerning the retinal artery occlusion event, fundus images were used for verification when available. Fundus images assisted in the evaluation of the presence of a retinal artery occlusion and, furthermore, the classification of the subtype. The diagnostic criteria used to determine the subtype included the presence of a cherry red spot, a visible embolus, and the proportion of the retina affected by ischemia caused by the occlusion. In cases where no images were accessible for validation, the verification was confirmed when two independent clinicians from the Department of Ophthalmology assessed and confirmed the occurrence of retinal artery occlusion events and if it was specified in the medical journal. If neither fundus photos or two-person verification was available, the validation was assessed based on the information available in the journal entry, including descriptions, visus, and symptoms.

Data were entered into RedCap (Research Electronic Data Capture, v11.0.3, Vanderbuilt University).

### 2.5. Statistical Analyses

The positive prediction value was calculated as the ratio between the number of verified diagnosis and the total number of diagnosis (N = 102). The Wilson score method was used to estimate the confidence interval for the positive prediction value. The validation was performed for retinal artery occlusion overall and at the subtype level (restricted to branch retinal artery occlusion and central retinal artery occlusion), since the diagnosis could be coded wrong at the subtype level. Furthermore, the positive prediction value was estimated for retinal artery occlusion stratified for the recorded ICD code group (branch retinal artery occlusion (DH342*) and central retinal artery occlusion (DH341*), excluding transient retinal artery occlusion (DH340*) due to an insufficient number of patients), age, sex, calendar year, and diagnosis type (primary or secondary). The strata were compared for homogeneity in the positive prediction value using Pearson’s ꭕ^2^ test.

All statistical analyses were conducted using Stata Statistical Software: Release 16 (StataCorp LP, College Station, TX, USA).

### 2.6. Ethics

This study is in compliance with the Declaration of Helsinki and the General Data Protection Regulation. The study is a part of the North Denmark Region’s record of processing activities. Research using data from Danish registers do not require approval from the Ethics Committee or patient consent.

## 3. Results

A total of 102 patients with recorded retinal artery occlusion diagnoses were identified and included from the register in the years 2017–2019. Of these, 90 were primary diagnoses, and the remaining 12 were secondary diagnoses. Five patients were registered with ICD-10 code DH340* for transient retinal artery occlusion (including subcodes), 52 were registered with ICD-10 code DH341* for central retinal artery occlusion (including subcodes), and 45 were registered with ICD-10 code DH342* for branch retinal artery occlusion (including subcodes). Fundus images or two-person verification was available for 73 (90%) of the verified patients.

The mean age of the included patients was 69.5 (SD: 12.0) years, and 33.3% of the population was female (Table 2). Out of the 102 included patients, a diagnosis of retinal artery occlusion was verified for 81 patients. This resulted in an overall positive prediction value of 79.4% (95% CI: 70.6–86.1%) for the total population of patients with recorded retinal artery occlusion diagnoses.

The positive prediction value was identified stratified according to age, sex, inclusion year, and diagnosis type. The positive prediction values of the stratified analyses ranged from 73.5 to 91.7%, with lower positive prediction values for the older age group, female patients, the most recent year of 2019, and for primary diagnoses. However, no statistically significant differences were identified among strata.

In the analysis validating the subtype diagnosis level, 71 diagnoses were verified, and the overall positive prediction value was 69.6% (95% CI: 60.1–77.7%), indicating that more errors were made at the subtype level compared to the overall disease level. In the stratified analyses at the subtype level, the positive prediction values ranged from 63.3 to 83.3%, which was lower than the validation at the overall level. No statistically significant differences were identified among the strata (see the positive prediction values for both the overall diagnosis analysis and subtype diagnosis analysis in Table 3).

Of the 102 patients with recorded retinal artery occlusion diagnoses, 21 were not verified as retinal artery occlusion events. Of these patients, eleven had a retinal vein occlusion, five experienced amarousis fugax, and the remaining five had other reasons for their visual impairments. In addition, for ten patients the wrong subtype diagnosis was registered.

Fundus photos provided valuable supplementary information for the subtype diagnosis validation and aided in verifying the diagnosis in most patients, complementing the information from the journal entries.

## 4. Discussion

In this validation study, the medical records of the patients with retinal artery occlusion showed an overall positive prediction value of 79.4%. The overall positive prediction value at the subtype diagnosis level was 69.6%, which suggests that the correct identification at the subtype diagnosis level is more challenging.

Validation studies help ensure the accuracy and reliability of the data recorded in the investigated registries. In Denmark, register-based studies rely on data routinely collected for administrative purposes. While these data sources can be valuable for research, they may contain errors or inaccuracies that can affect the validity of study results. In this validation study, we compared data in registries to data collected through medical records supplemented by information from physical examinations and imaging. Thereby, the agreement between the registrations and the medical records and the accuracy and completeness of the registry data can be determined.

We assessed other validation studies to determine what an acceptable positive prediction value for a population would be for use in research. A positive predictive value of 80% or above is generally considered a high positive predictive value. A value of 60% or less would be considered a low positive predictive value [18,19,20]. Therefore, the positive predictive value for retinal artery occlusion of 79.4% is moderate–high, and the positive predictive value at the subtype diagnosis level of 69.6% is moderate. A moderate–high positive predictive value is considered acceptable for use in research. A moderate positive predictive value is also acceptable for use in research, however a moderate positive predictive value should be considered when interpreting analyses and may be a limitation of a study.

We did not identify noticeable differences between the overall positive prediction value and the stratified positive prediction values or among the investigated strata either in the validation at retinal artery occlusion diagnosis level or at the subtype diagnosis level. The positive predictive values were moderate or high for all investigated subgroups, with smaller variations among subgroups. This suggests generally reliable recordings of retinal artery occlusion in the Danish National Patient Registry.

No previous validation studies on retinal artery occlusion diagnoses in the Danish National Patient Registry were identified, preventing comparison. However, other cardiovascular diagnoses have been validated in the Danish National Patient Registry, with overall positive prediction values ranging from 64% to 100% [19]. This suggests that the positive prediction value for retinal artery occlusion is comparable with other cardiovascular disease diagnoses in the Danish National Patient Registry, showing an overall reliable recording of diagnoses in the register. Another study validated the diagnosis codes for the Charlson comorbidities and found positive predictive values ranging from 82 to 100%, and these positive predictive values were high compared to other validation studies [20]. The positive predictive values have been investigated for the diagnosis codes for stroke, where a positive predictive value of 79.3% was identified for overall stroke diagnoses [18]. This was comparable with the overall positive predictive value identified in this study. These studies were all conducted in the Danish National Patient Registry, which indicates a generally acceptable validity of the register.

The ICD-10 codes for ophthalmologic diseases are not precise for the variety of possible diseases, making specific recording challenging. For instance, no specific ICD-10 code is available for cilioretinal occlusion, and it is generally coded as a branch retinal artery occlusion. To evaluate the subtype diagnosis, fundus photos are used as ground truth, which are a valuable supplement to the journal entries. Fundus photos supported the verification of the diagnosis for the majority of the patients. In the few cases where fundus photos and two-person verification were not available, we had to rely on the journal entries. In these cases, the visus of the affected patients were a good indicator of the subtype. Patients with central retinal artery occlusion will often have a more affected vision compared to patients with branch retinal artery occlusion.

There were 21 diagnoses that were not verified as retinal artery occlusion events. Of these patients, eleven had a retinal vein occlusion, five experienced amarousis fugax (where retinal artery occlusion could not be determined as the cause), and the remaining five had other reasons for their visual impairments. The findings suggest that a common error in diagnostic registration involves erroneously categorizing other eye diseases as retinal artery occlusion. The ICD-10 code for retinal vein occlusion, denoted as DH348, is described as another vascular occlusion of the retina. Although more specific ICD-10 codes specify retinal vein occlusion, the descriptive nature of these codes can potentially contribute to errors within a busy healthcare system. Furthermore, the subgroups for retinal artery occlusion and retinal vein occlusion share similarities in their descriptive terms, including branch and central occlusions, further increasing the likelihood of confusion between the two conditions. Amarousis fugax was the cause of some of the incorrect registrations for retinal artery occlusion. Amarousis fugax may be caused by a transient occlusion and can only be diagnosed when the patient regains vision; however, other vision-impairing events may cause amarousis fugax as well. Therefore, transient retinal artery occlusion and amarousis fugax should not be used interchangeably when registering temporary vision loss.

In the subgroup analyses, 71 of the 102 patients with retinal artery occlusion were verified. This was a difference of ten compared to the main analysis, which corresponds to a positive predictive value of 69.6%. A common characteristic of central retinal artery occlusion is a cherry red spot, which is not present in branch retinal artery occlusions. Therefore, it should be possible to distinguish the subtypes from each other, which indicates that an error may occur during the registration. In the ICD-10 coding system, the classification of a central retinal artery occlusion is unambiguous, as it is assigned the code DH341, with a clear description as “occlusion arteriae centralis retinae”. However, the coding for branch retinal artery occlusion is denoted as DH342, which has a broader definition of *another type of retinal artery occlusion*, lacking specific description for branch retinal artery occlusion. This discrepancy in the specificity of the ICD-10 codes could cause some of the incorrect registrations of the subtype diagnosis for retinal artery occlusion, which could lead to incorrect coding for both subtypes. However, the descriptive nature of the ICD-10 codes may cause more branch retinal artery occlusions to be registered as central retinal artery occlusions, which is supported by the results of this study.

In general, the predominant error observed in this study was the uncertainty regarding the precise registration code to be used. The most common mistakes occurred between retinal artery occlusion and retinal vein occlusion, as well as between central retinal artery occlusion and branch retinal artery occlusion. Preventing these errors may be challenging because of the demanding nature of a busy healthcare system, where time constraints may hinder the selection of the most accurate or specific registration code. These mistakes are likely attributed, to a significant extent, to factors such as the oversight and confusion inherent in a fast-paced healthcare environment, where there may be limited time available for registering diagnoses.

This study validated all diagnoses of retinal artery occlusions made by the Department of Ophthalmology in Northern Jutland in Denmark in the years 2017, 2018, and 2019. We were not in a position to verify that this validation study was representative for previous years. Furthermore, we only validated the existence of a retinal artery occlusion diagnoses, not the time aspect of the event. This was mainly an issue for the secondary diagnoses, where the diagnosis, in some cases, were given because of an earlier event. Eight patients were excluded because either ophthalmologic journal entries or fundus imaging confirmed that the retinal artery occlusion event occurred before 2018. This could be due to the previous use of a different code for retinal artery occlusion. There were some differences in the number of cases occurring between the included years, varying from 23 cases in 2018 to 44 cases in 2019. This could potentially influence the results. However, we did not identify any statistically significant differences among the included years, which suggests that the varying number of yearly cases did not significantly influence the results.

It is not possible to calculate the sensitivity, specificity, and negative predictive values based on the information available in this study. This would only be possible if we had access to the false negatives and the true negatives, which would require assessing all patients in the registry.

Another limitation of this study was the number of included patients. Retinal artery occlusion is a relatively rare disease, resulting in a low number of potential patients. We wanted to include recent years to ensure the results were representative for the current recording of retinal artery occlusions in the registry. We included all patients with a diagnosis of retinal artery occlusion between 2017 and 2019, resulting in 102 patients and assessed that this would be sufficient to assess the positive prediction value for this diagnosis.

Data were restricted to the North Denmark Region, which could be a limitation to the study. There may be some differences between the population from the North Denmark Region and other regions in Denmark. However, these differences in the population could influence the number of patients and when events are diagnosed. However, this was not part of the validation in this study, and the Danish healthcare system is generally uniform across all regions, which is ensured by the tax-supported free healthcare that is accessible to all, as described in Section 2.

## 5. Conclusions

In conclusion, our validation study provides evidence of an acceptable accuracy and reliability of the recording of retinal artery occlusion diagnoses in the Danish National Patient Registry. We demonstrated that the recorded diagnoses are valid and can be used in epidemiological studies.

## Figures and Tables

**Table 1 jpm-13-00970-t001:** The specific ICD-10 codes for retinal artery occlusion.

ICD-10 Code	Description
DH340	Transient retinal artery occlusion
DH340A	Ischemia transitoria retinae
DH341	Occlusion arteriae centralis retinae
DH341A	Embolia arteriae centralis retinae
DH341B	Thrombosis arteriae centralis retinae
DH342	Other retinal artery occlusions
DH342A	Cholesterol plaques retinae
DH342B	Microembolus retinae
DH342C	Occlusion arteriae retinae partialis

**Table 2 jpm-13-00970-t002:** Baseline characteristics of the population comprising patients with retinal artery occlusion.

Characteristic	Population
N	102
Age, mean (SD)	69.5 (12.0)
Female, % (N)	33.3 (34)
Central retinal artery occlusion, % (N)	51 (52)
Branch retinal artery occlusion, % (N)	44 (45)
Transient retinal artery occlusion, % (N)	5 (5)

**Table 3 jpm-13-00970-t003:** Validation of the overall and subtype diagnoses of retinal artery occlusion.

			Overall Diagnosis			Subtype Diagnosis		
Strata		No. of Patients	Verified	PPV (95% CI)	*p*-Value	Verified	PPV (95% CI)	*p*-Value
Retinal artery occlusion		102	81	79.4 (70.6–86.1)		71	69.6 (60.1–77.7)	
ICD code group					0.39			0.81
	BRAO	45	35	77.8 (63.0–88.8)		33	73.3 (58.1–85.4)	
	CRAO	52	44	84.6 (71.9–93.1)		37	71.2 (56.9–82.9)	
Age group (years)					0.84			0.38
	24–65	29	24	82.8 (64.2–94.2)		23	79.3 (60.3–92.0)	
	66–75	43	34	79.1 (64.0–90.0)		29	67.4 (51.5–80.9)	
	76–94	30	23	76.7 (57.7–90.1)		19	63.3 (43.9–80.1)	
Sex					0.30			0.88
	Male	68	56	82.4 (71.2–90.5)		47	69.1 (56.7–79.8)	
	Female	34	25	73.5 (55.6–87.1)		24	70.6 (52.5–84.9)	
Inclusion year					0.50			0.74
	2017	35	30	85.7 (69.7–95.1)		26	74.3 (56.7–87.5)	
	2018	23	17	73.9 (51.6–89.8)		15	65.2 (42.7–83.6)	
	2019	44	34	77.3 (62.2–88.5)		30	68.2 (52.4–81.4)	
Diagnosis type					0.26			0.27
	Primary	90	70	77.8 (67.8–85.9)		61	67.8 (57.1–77.2)	
	Secondary	12	11	91.7 (61.5–99.8)		10	83.3 (51.6–97.9)	

No. of patients is the total number of patients in the specific strata. Verified is defined as the number of patients where the registered ICD-10 code matched the medical record. The ratio between the number of verified and total number of patients was calculated as the positive prediction value and homogeneity among the strata assessed through the *p*-value. PPV, positive prediction value; BRAO, branch retinal artery occlusion; CRAO, central retinal artery occlusion.

## Data Availability

According to Danish law, individual-level data may not be deposited.

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
