# Peer review of "Positive Prediction Value of Retinal Artery Occlusion Diagnoses in the Danish National Patient Registry: A Validation Study"

_jpm, 2023, doi:10.3390/jpm13060970_

Round 1
Reviewer 1 Report
I would make the abstract extra clear. I would change the synthax, but I think that the article per se is otherwise well written. I would stress what the novelty is from this study.
Congrats on the paper.
Good
Author Response
Thank you for reviewing our paper and for your comments.
I would make the abstract extra clear. I would change the synthax, but I think that the article per se is otherwise well written. I would stress what the novelty is from this study.
Response: Thank you for the comments. We have updated the abstract and tried to make it more clear. Furthermore, we have expanded the paper in general and specified some of the points.
Reviewer 2 Report
This study by Marie et al. proposes the notion that the validity of the RAO diagnosis was acceptable in the Danish National Registry. The manuscript is interesting and generally well-written, and the presentation of the content in the table is good. However, I have some concerns that the authors should take into account (see detailed suggestions below).
1. Table 1 was not provided. Confusingly, there is only one table in the manuscript named Table 2.
2. Table 1 is hard to understand because of the lack of an explanation for each column. There should have been an explanation for the left and right columns (verified, PPV, and p-values). Moreover, the sum of BRAO and CRAO subtypes (35+44 and 33+37) did not match the top row data (81 and 71, respectively).
3. They reviewed fundus photographs and verified if the patients were diagnosed with CRAO, BRAO, or another diagnosis. The specific diagnostic criteria of fundus photographs for CRAO and BRAO should be described in the Methods section.
4. The authors insisted that the Registry has an acceptable validity for RAO diagnosis because its PPV is 69.6. However, considering that this study is based on a relatively small group of the population, this statement can be controversial. A discussion about the PPV for other diseases in the Danish Registry is essential for strengthening their opinion.
5. Adequate abbreviations should be reviewed (e.g., positive prediction value was abbreviated in line 49 and then abbreviated again in line 113).
Moderate editing of English language is required.
Author Response
Thank you for your comments. Your comments were well-considered and has helped to improve the manuscript.
- Table 1 was not provided. Confusingly, there is only one table in the manuscript named Table 2.
Response: Thank you for this comment. We have added two additional tables. One of the assessed ICD-10 codes and a table 1.
- Table 1 is hard to understand because of the lack of an explanation for each column. There should have been an explanation for the left and right columns (verified, PPV, and p-values). Moreover, the sum of BRAO and CRAO subtypes (35+44 and 33+37) did not match the top row data (81 and 71, respectively).
Response: This is a great point. We have added a text explaining the table. Furthermore, we have noted in the methods that subtype only included branch and central retinal artery occlusion and that transient retinal artery occlusion were excluded in the stratified analysis due to low prevalence.
- They reviewed fundus photographs and verified if the patients were diagnosed with CRAO, BRAO, or another diagnosis. The specific diagnostic criteria of fundus photographs for CRAO and BRAO should be described in the Methods section.
Response: Thank you for the comment. This is a great point and we agree. We have added the diagnostic criteria assessed when determining the subtype in the methods section.
- The authors insisted that the Registry has an acceptable validity for RAO diagnosis because its PPV is 69.6. However, considering that this study is based on a relatively small group of the population, this statement can be controversial. A discussion about the PPV for other diseases in the Danish Registry is essential for strengthening their opinion.
Response: Thank you for this comment. In the discussion we have added a section discussing what a high, moderate, and low positive prediction value is.
- Adequate abbreviations should be reviewed (e.g., positive prediction value was abbreviated in line 49 and then abbreviated again in line 113).
Response: Thank you, we have corrected the abbreviations.
Round 2
Reviewer 2 Report
All suggested questions are answered properly.